# MVFL: Multivariate Vertical Federated Learning for Time-Series Forecasting

## Abstract

Extending multivariate time series forecasting to resource-limited devices is a critical demand for real applications, especially with the advancements in IoT technologies. A common scenario is where the variates are distributed vertically on different devices and each device needs to do local forecasting. This paper studies a resource-efficient solution for this scenario based on vertical federated learning (VFL). Prior VFL frameworks are designed for situations where only one party holds the labels and would struggle to meet the demand of the targeted scenario, as storage resources usage would increase dramatically with the number of devices. Going beyond VFL, we design multivariate vertical federated learning (MVFL) as a novel federated learning framework, where we separate communication features and local features in an embedded feature space. This design enables MVFL to utilize storage and communication resources more efficiently by eliminating the redundant models. MVFL outperforms VFL approaches in both efficiency and accuracy. On four real-world benchmarks, compared to VFL, when the storage resources are equally utilized, MVFL yields a 12.1% relative improvement on loss with a 43% relative improvement on communication resources usage. Even when both MVFL and VFL employ the same main model size, MVFL achieves a 75% reduction in storage resources compared to VFL while maintaining the loss at the same level of VFL.

## 1 Introduction

Time series forecasting on end devices is widely used in many aspects such as traffic or energy planning, weather and disease propagation forecasting, and is gaining increasing importance with the development of new methods such as digital twins (Fahim et al. (2022)), multi-modal models (Hayat et al. (2022)), IoT technologies (Bauer et al. (2021)) and so on. In these real-world scenarios, there is always a critical need to gather useful information from different devices in order to make predictions. The devices may hold distinct features and heterogeneous data (Lin et al. (2024), Yang et al. (2019a)). Furthermore, ensuring the privacy of each device is essential for security purposes (Tawalbeh et al. (2020)). In this paper, we focus on time series forecasting of multivariate data with the variates distributed across different devices. Recent studies has primarily employed the federated learning (FL) method to to address privacy concerns (Gosselin et al. (2022)), with vertical federated learning (VFL) emerging as a particularly effective method for handling vertically partitioned data (Yang et al. (2019b)).

However, the forecasting task becomes extremely challenging under the scenario where each device is expected to make predictions on its own data: The VFL approach would demand significant computational and storage resources, as each device needs to build separate models for all other devices (see Section 3). Compounding the issue, many devices in this targeted scenario are inherently resource-limited due to factors such as low processing power, limited memory, and energy constraints (Imteaj et al. (2021)). These limitations arise from the need for devices to be cost-effective and energy-efficient, which is particularly crucial in IoT applications where large numbers of devices are deployed in the field. Consequently, this resource scarcity leads to challenges in handling complex tasks like forecasting, as devices struggle to balance storage demands with their available resources.

To address the aforementioned challenge, we focus on the *useful information* extracted from each device. We notice that some information extracted from one device of the targeted scenario might be beneficial for many other devices as well. For instance, in weather forecasting, a rising temperature trend not only aids in predicting future temperatures but can also contribute to forecasting wind direction and rainfall. Constructing separate models to extract similar information would thus be inefficient. Therefore, in this paper, we seek a method to merge the separate models on a given device in the VFL approach into one integrated model in order to save storage resources without sacrificing accuracy. In typical VFL settings, the passive parties (devices without labels) send their extracted information to the active party (the device with labels). For any device in our scenario, we try to extract *communication features* from an embedded feature space of the local model to facilitate interaction with other devices..

Based on the above motivations, we propose an original **Multivariate Vertical Federated Learning (MVFL)** in place of VFL for forecasts of multivariate time series with the variables verically distributed across different devices. MVFL retains the core principles of VFL but refines the overall architecture into a more compact and efficient structure. This design allows each device to maintain only one model for the prediction tasks. MVFL also introduces the concepts of *communication features* and *local features* which allows us to distinguish between information that is valuable for communication with other devices and information that is useful solely for the local device. MVFL has been demonstrated to achieve lower loss and improved efficiency in storage and communication resources compared to the traditional VFL approach. Our contributions are summarized as follows:

- We identify the challenges that classical federated learning approaches face in a new research scenario of multivariate time series forecasting, where the variables of the dataset are vertically distributed across different devices, and each device must predict its own future data.

- We propose MVFL with a compact structure and the separation of *communication features* and *local features*. We adopt an approach *a posteriori* rather than *a priori*, meaning the features evolve throughout the training process rather than being predefined.

- Across four real-world benchmarks, MVFL achieves a loss that is 87.9% of that of VFL with a 83% storage resources usage and a 57% communication resources usage. In more extreme situations, while maintaining the loss at the same level, MVFL could reduce the storage resources usage to only 25% of that of VFL.

## 2 RELATED WORKS

### 2.1 FEDERATED LEARNING & VERTICAL FEDERATED LEARNING

Federated Learning (FL) has emerged as a powerful paradigm to enable decentralized machine learning, where devices collaboratively train a model without sharing raw data, thereby preserving privacy (Mammen (2021)). Early foundational works (McMahan et al. (2017)) introduced a framework that reduces communication costs by averaging model updates across devices. Since then, FL has been extensively applied to domains where data privacy is crucial (Truex et al. (2019), Mothukuri et al. (2021)), including healthcare, financial services, and IoT (Kairouz et al. (2021), Bonawitz (2019)). However, traditional FL techniques primarily assume horizontally partitioned data, where different devices hold different samples of the same features.

There are FL approaches concentrating on multi-modality (Che et al. (2023), Peng et al. (2024), Sundar et al. (2024)) that could provide useful insights about vertically distributed data. However, For all horizontal FL-based approaches, once all the models have finished training, they could do their tasks solely relying on local data and model while in the targeted scenario, a device would need the simultaneous data (or at least the extracted information of the data) from other devices in order to perform local forecasting.

In contrast, Vertical Federated Learning (VFL) is designed for scenarios where different parties hold different features for the same set of data instances (Romanini et al. (2021), Wu et al. (2020), Gu et al. (2021)). VFL has proved to be effective in domains like finance and healthcare, where features are naturally distributed across different organizations (Liu et al. (2024), Chen et al. (2022)). Yang et al. (2019b) presented a comprehensive overview of VFL methodologies and challenges, highlighting

issues such as communication efficiency, model design, and privacy concerns when multiple parties collaborate under a vertically partitioned data framework.

Nevertheless, current VFL frameworks are primarily designed for scenarios where only one party holds the labels (Chen et al. (2020), Zhang et al. (2021), Gu et al. (2020)), incompetent when dealing with the targeted scenario. Xia et al. (2021) is the only work we could find that deals with a similar scenario with the targeted scenario, where labels are horizontally partitioned and the parties only hold partial labels. However, in the targeted scenario, both labels and data are vertically distributed. Besides, in Xia et al. (2021), there is still the distinction of active parties, passive parties and a collaborator, since the framework isn't specially designed for time series forecasting tasks. As a result, none of the parties are equipped with sufficient labels to perform a complete local training, which is not the case in the targeted scenario, as local future data are natural labels for time series forecasting tasks.

## 2.2 MULTIVARIATE TIME SERIES FORECASTING

Multivariate time series forecasting is becoming increasingly important in IoT contexts (Papastefanopoulos et al. (2023)), where devices handle tasks like predicting energy consumption, traffic flow, and weather patterns (Papastefanopoulos et al. (2023), Han et al. (2021), Bitencourt & Guimarães (2021)). Recent approaches include transformer-based models such as Zhou et al. (2021) and other approaches (Wu et al. (2021), Yu et al. (2024), Oreshkin et al. (2019)) that further integrates decomposition methods. However, many of these devices are resource-limited, with constraints on processing power, memory, and energy (Imteaj et al. (2021)). Federated Learning (FL) has been applied to IoT systems to mitigate the challenges of decentralized data processing (Imteaj et al. (2022)). Specially, Chen et al. (2023) introduces a prompt learning mechanism to accommodate the communication and computational constraints of low-resource sensors. However, most researches has not adequately addressed how to handle multivariate forecasting with vertically distributed variates.

Efforts have been made to reduce FL's resource footprint (Imteaj & Amini (2021), Zakariyya et al. (2022), Shen et al. (2020)), but those methods do not address the redundancy issues inherent in VFL (see Section 3) for large-scale multivariate time series forecasting with vertically partitioned data, where each device needs to build separate models for any other variate. The lack of research addressing these specific challenges represents a critical gap in the literature.

**To the best of our knowledge, our study addresses this gap by proposing a novel framework, Multivariate Vertical Federated Learning (MVFL), which is tailored for the targeted scenario.** Unlike previous methods, MVFL improves both storage efficiency and forecasting accuracy by eliminating redundant models and efficiently utilizing device resources.

## 3 LIMITATIONS OF VFL

In a typical VFL setting with $n$ devices (Zou et al. (2023)), each device holds its local private data $X_i$, where $i \in \{1, 2, \ldots, n\}$. Only one device (we could denote it as device $k \in \{1, 2, \ldots, n\}$) contains the labels $Y_k$ and is referred to as active party. The active party is equipped with a model $M_k$ with parameters $\theta_k$. The other devices are called passive parties. For those passive parties, each device $i$ also maintains a local model $M_i$ with parameters $\theta_i$. During the forward propagation, the device $i$ would compute the communicated features $C_i = M_i(X_i)$ and send them to the active party. The active party would aggregate the communicated features from passive parties with its own local data (a common practice is concatenation) and compute $\hat{Y}_k = M_k(aggregation(X_i))$. Define the loss as $\ell(\hat{Y}_k, Y_k)$, then during the back propagation, the active party would compute $g_k = \frac{\partial \ell}{\partial \theta_k}$ in order to update its own model. It would also compute $g_{i,c} = \frac{\partial \ell}{\partial C_i}$ and send them back separately to the passive parties. The passive parties would then compute $g_i = g_{i,c} \cdot \frac{\partial C_i}{\partial \theta_i}$ in order to update the local models.

However, in our targeted scenario, all $n$ devices need to do forecasting for the local variates using useful information extracted from other devices, which means that each party serves as both an active and a passive party simultaneously. Specifically, for any device $i$, apart from a model intended for doing local forecasting, which we refer to as the main model, it should also maintain $(n-1)$ models

to extract information from its own raw data in order to provide useful information for all other devices, which we refer to as exchange models. Consequently, as the number of devices increases, the total number of models that need to be maintained also grows, leading to significant storage and computational overhead. This reality motivates us to propose a novel federated learning framework tailored for this scenario.

# 4 MULTIVARIATE VERTICAL FEDERATED LEARNING

With the above discussions, we could now renovate the typical VFL framework for the targeted scenario. We name the proposed method the multivariate vertical federated learning framework (MVFL).

Concretely, Suppose that there is a multivariate time series dataset $D \in \mathbb{R}^{l \times n}$, where $l$ denotes the size of any variate in this dataset and $n$ denotes the number of variates in this dataset, Then the targeted scenario is where the variates are distributed exactly on $n$ devices, and any local dataset could be denoted as $D_i \in \mathbb{R}^l, i \in \{1, 2, \ldots, n\}$. The objective of a device is to predict the most probable length-$O$ series in the future given the past length-$I$ series of its own variate combined with information sent from other devices. We note the input $X_{iI} \subseteq D_i$ and output $X_{iO} \subseteq D_i$.

A simple illustrative example of MVFL for three devices is as shown in Figure 1.

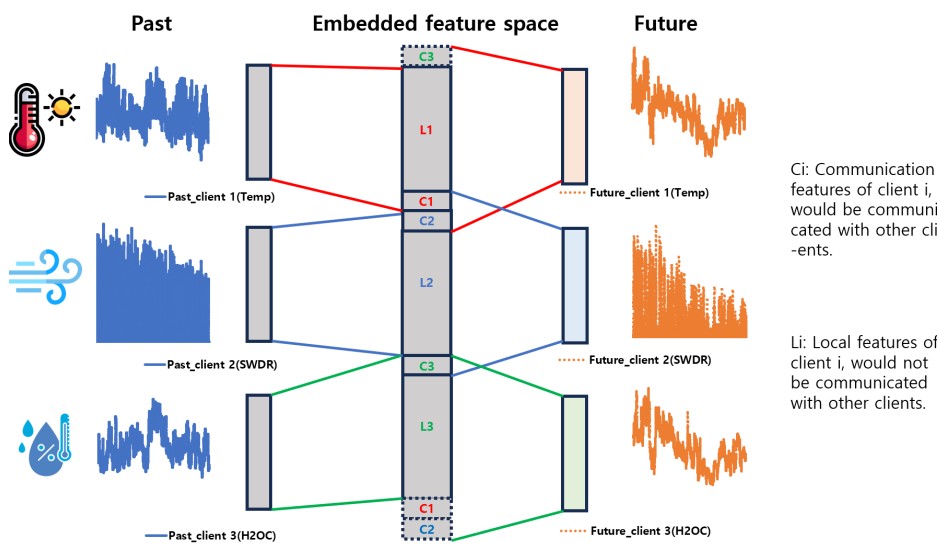

Figure 1: Illustration of MVFL with a simple example of three devices separately holding data of temperature, wind, and h2o density. They exchange communication features in order to make predictions of own data.

## 4.1 FORWARD PROPAGATION

In MVFL, a device only needs to maintain one model: its own forecasting model. What are sent to other devices are simply the communication features on one of the hidden layers of its own model.

Concretely, denote the model to maintain for device $i$ as $M_i$. $M_i$ could be thought of as consisted of two parts: $M_{i,1}$ and $M_{i,2}$. We note $C_i$ and $L_i$ separately for communication features and local features.The details of the forward propagation could be described as follows:

$$C_i, L_i = M_{i,1}(X_{iI}) \tag{1}$$

$$C_{i,other} = \text{concatenation}(C_j, j \text{ from 1 to } n, j \neq i) \tag{2}$$

$$\hat{X}_{iO} = M_{i,2}(C_i, L_i, C_{i,other}) \tag{3}$$

Specifically, the features $C_{i,other}$ concatenates the communication features from all other devices. This is feasible because during the forward propagation, communication features from all the devices would be sent to a trusted server, and the server would do the concatenation and send those concatenated features (Concretely, $C_{i,other}$ for device $i$) separately to the devices.

---

**Algorithm 1** Behavior of client $i$ during a communication round

---

1: $\alpha \leftarrow$ Learning rate
2: $C_i, L_i \leftarrow M_{i,1}(X_{iI})$
3: Send $C_i$ to server
4: Receive $C_{i,other} = \text{concatenation}(C_j, j \text{ from } 1 \text{ to } n, j \neq i)$ from server
5: $\hat{X}_{iO} \leftarrow M_{i,2}(C_i, L_i, C_{i,other})$
6: $\ell_i \leftarrow Lossfunction(\hat{X}_{iO}, X_{iO})$
7: $g_i^1, g_i^2, g_i^o \leftarrow \frac{\partial \ell_i}{\partial \theta_{i,1}}, \frac{\partial \ell_i}{\partial \theta_{i,2}}, \frac{\partial \ell_i}{\partial C_{i,other}}$
8: Send $g_i^o$ to server
9: Receive $g_i^c$ from the server.
10: $g_i^3 \leftarrow g_i^c \cdot \frac{\partial C_i}{\partial \theta_{i,1}}$
11: $\theta_{i,1}, \theta_{i,2} \leftarrow \theta_{i,1} - \alpha \cdot (g_i^1 + g_i^3), \theta_{i,2} - \alpha \cdot g_i^2$

---

**Algorithm 2** Behavior of the server during a communication round

---

1: **for** $i \leftarrow 1$ **to** $n$ **do**
2:    Receive $C_i$ from device $i$
3: **end for**
4: **for** $i \leftarrow 1$ **to** $n$ **do**
5:    Send $C_{i,other} = \text{concatenation}(C_j, j \text{ from } 1 \text{ to } n, j \neq i)$ to device $i$
6: **end for**
7: **for** $i \leftarrow 1$ **to** $n$ **do**
8:    Receive $g_i^o = \text{concatenation}(g_{i,j}, j \text{ from } 1 \text{ to } n, j \neq i)$ from device $i$
9: **end for**
10: **for** $i \leftarrow 1$ **to** $n$ **do**
11:    Send $g_i^c = \sum_{j \neq i} g_{j,i}$ to device $i$
12: **end for**

---

## 4.2 BACK PROPAGATION

The back propagation of MVFL is a modified version of the classical VFL back propagation procedure. Concretely, for a client $i$, we note $\theta_{i,1}$ as the model parameters for $M_{i,1}$ and $\theta_{i,2}$ the model parameters for $M_{i,2}$. We note also the loss of the overall model as $\ell_i(\hat{X}_{iO}, X_{iO})$, then during back propagation, the following gradients are calculated:

$$g_i^1 = \frac{\partial \ell_i}{\partial \theta_{i,1}} \tag{4}$$

$$g_i^2 = \frac{\partial \ell_i}{\partial \theta_{i,2}} \tag{5}$$

$$g_i^o = \frac{\partial \ell_i}{\partial C_{i,other}} \tag{6}$$

While the gradients $g_i^1$ and $g_i^2$ stay local, the client would send $g_i^o$ to the server. We notice that $g_i^o = \text{concatenation}(g_{i,j}, j \text{ from } 1 \text{ to } n, j \neq i)$ with $g_{i,j}$ denoting the gradients from client $i$ for communication features of client $j$. After having collected the gradients from all clients, the server would calculate:

$$g_i^c = \sum_{j \neq i} g_{j,i} \tag{7}$$

and send this gradient to client $i$. Consequently, client $i$ could calculate another gradient:

$$g_i^3 = g_i^c \cdot \frac{\partial C_i}{\partial \theta_{i,1}} = \sum_{j \neq i} \frac{\partial \ell_j}{\partial \theta_{i,1}} \tag{8}$$

Finally, the client $i$ could update its gradients for $\theta_{i,1}$ using $(g_i^1 + g_i^3)$ and the gradients for $\theta_{i,2}$ using $g_i^2$.

The detailed algorithms for any client and the server are as shown in Algorithm 1 and Algorithm 2.

### 4.3 SEPARATION OF COMMUNICATION AND LOCAL FEATURES

The key insight behind this solution is that among all the features extracted by the $(n-1)$ exchange models in the classical VFL solution, some features may be redundant. Besides, a feature extracted from a specific variate that is useful for predicting its own future data can also be beneficial for forecasting other variates. Therefore, maintaining separate exchange models can be inefficient. Instead, we should directly examine the embedded feature space of the local main model to identify the features that are useful for forecasting the variates on other devices while retaining the remaining features for local use.

A key challenge in implementing this insight is how to distinguish between communication features and local features. Our solution is rather an approach *a posteriori* than *a priori*. Concretely, at the beginning of the training process, the only distinction between communication features and local features lies in their positions within the embedded feature space of the local model. However, for any round of training, for a device $i$, the gradients for the communication features $C_i$ could be expressed as

$$\sum_{j \in \{1, 2, \ldots, n\}} \frac{\partial \ell_j}{\partial C_j} \tag{9}$$

while the gradients for the local features $L_i$ could be expressed as $\frac{\partial \ell_i}{\partial L_i}$. Consequently, As the training process progresses, the communication features and local features will gradually diverge and converge into their respective identities as suggested by their names.

### 4.4 COMMUNICATION RESOURCES USAGE

to compare the communication resources usage of VFL and MVFL, we could fix the size of communication features of any client of MVFL as $e$. Meanwhile, for VFL, we fix the size of the exchanged features of any client to any other client also as $e$.

In MVFL framework, for every round of training, for a client $i$, what are communicated with the server include:

- What are sent to the server: the communication features $C_i$ obtained from local model of size $e$ and the gradients $g_i^o$ for the communication features of other clients of size $e \cdot (n-1)$.

- What are received from the server: The communication features form all other clients $C_{i,other}$ of size $e \cdot (n-1)$ and the gradients for own communication features $g_i^c$ of size $e$.

The overall communication resource usage of MVFL is thus of size $\mathbf{2n \cdot e}$.

In VFL framework, for every round of training, for a client $i$, what are communicated with the server include:

- What are sent to the server: the communicated features $C_i$ obtained from local model of size $e \cdot (n-1)$ and the gradients $g_i^o$ for the communicated features of other clients of size $e \cdot (n-1)$.

- What are received from the server: The communicated features form all other clients $C_{i,other}$ of size $e \cdot (n-1)$ and the gradients for own communicated features $g_i^c$ of size $e \cdot (n-1)$.

The overall communication resource usage of VFL is thus of size $(\mathbf{4n} - \mathbf{4}) \cdot \mathbf{e}$.

We remark that under the same $e$, the communication resource usage of MVFL is far below that of VFL. Concretely, ith the increase of the number of devices, the ratio of communication resource usage of MVFL over VFL would tend to $50\%$.

## 5 EXPERIMENTS

We evaluate the proposed MVFL by comparing it to the classical VFL approach on four real-world benchmarks, covering the mainstream time series applications of energy, weather and economics.

### 5.1 DATASETS

The datasets that we use are *ETTm*, *ETTh*, *weather* and *exchange*. All are multivariate datasets. *ETTm* and *ETTh* each has 7 variates. The *weather* dataset has 21 variates, the *exchange* dataset has 8 variates. All data are normalized using a standard scaler which ensures the mean value of a variate is 0 and the standard deviation is 1.

Here is a brief description (Wu et al. (2021)) of the datasets used: (1) *ETT* (Zhou et al. (2021)) dataset contains the data collected from electricity transformers, including load and oil temperature between July 2016 and July 2018. For *ETTh*, the data are recorded every 1 hour. For *ETTm*, the data are recorded every 15 minutes. (2) *weather*[1] is recorded every 10 minutes for 2023 whole year, which contains 21 meteorological indicators, such as air temperature, humidity, etc. (3) *exchange* (Lai et al. (2018)) records the daily exchange rates of eight different countries ranging from 1990 to 2016.

### 5.2 IMPLEMENTATION DETAILS

Our models are trained with the L2 loss ($l(a,b) = (a-b)^2$), using the ADAM optimizer with an initial learning rate of 0.001. Batch size is set to 32. The training process has its learning rate decayed if there is a stagnation of loss. All experiments are implemented in PyTorch and conducted on a single NVIDIA GPUs. We use Mean Squared Error (MSE) as the core metric to compare performances.

Both the MVFL model and the VFL main model contain 3 hidden layers. For MVFL, the 2nd hidden layer serves as the embedded feature space to obtain the *communication features*. To maximize control over variables and ensure best performances, for VFL, the exchange models all have 1 hidden layer, and the exchanged features obtained from other devices are concatenated with own past data. The storage resources usage are in KB and are obtained by saving all the models locally and adding the sizes of the models saved. The communication resources usage are in bytes and are calculated as shown in Section 4.4.

### 5.3 MAIN RESULTS

To compare performances under long and short horizons, we fix the output length and evaluate models with two input lengths: 96, 24. We set the communication features size to 6 and do the VFL comparison experiments with the same exchanged features sizes.

Note that MVFL could greatly reduce storage and communication resources usage compared to VFL. Consequently, we focus on three indexes for the experiments: loss, communication resources usage and storage resources usage. The first comparison experiments, denoted as MVFL1, control the same storage resources usage, where the size of MVFL models are extended in order to approach the storage resources usage of VFL. The second comparison experiments, denoted as case MVFL2, control the same main model size, where MVFL models are not extended and have the same size of the VFL main models. The results are as shown in Table 1.

We remark that MVFL achieves consistent advantageous performance in all benchmarks and both input size settings. Especially, under the input-96 setting for the exchange dataset, compared to the

---

[1]https://www.bgc-jena.mpg.de/wetter/

Table 1: Comparison of MVFL and VFL under different input sizes. The loss is MSE loss, the storage is the sum of model sizes and is in KB. The communication is the size of communicated information per training round and is in bytes. For all three criteria, a lower value indicates a better performance.

| Datasets | | ETTm | | ETTh | | weather | | exchange | |
|---|---|---|---|---|---|---|---|---|---|
| Input size | | 96 | 24 | 96 | 24 | 96 | 24 | 96 | 24 |
| MVFL1 | Loss | **1.21** | **1.72** | **2.14** | **2.39** | 4.10 | **3.84** | **0.23** | **0.20** |
| | Storage | 103.8 | 40.3 | 103.8 | 40.3 | 290.4 | 105.7 | 114.8 | 45.5 |
| | Communication | **168** | **168** | **168** | **168** | **504** | **504** | **192** | **192** |
| VFL | Loss | 1.31 | 1.74 | 2.28 | 2.51 | 4.29 | 3.87 | 0.36 | 0.31 |
| | Storage | 114.5 | 51.5 | 114.5 | 51.5 | 330.6 | 141.6 | 130.0 | 58.0 |
| | Communication | 288 | 288 | 288 | 288 | 960 | 960 | 336 | 336 |
| MVFL2 | Loss | 1.28 | 1.79 | 2.33 | 2.52 | **4.02** | 3.90 | 0.29 | 0.27 |
| | Storage | **26.5** | **17.5** | **26.5** | **17.5** | **40.0** | **28.0** | **27.2** | **18.2** |
| | Communication | 168 | 168 | 168 | 168 | 504 | 504 | 192 | 192 |

Table 2: Loss of MVFL and VFL over different exchange sizes under the input-96 and input-24 setting

| Datasets | ETTm | | | ETTh | | | weather | | | exchange | | |
|---|---|---|---|---|---|---|---|---|---|---|---|---|
| exchange size | 3 | 6 | 9 | 3 | 6 | 9 | 3 | 6 | 9 | 3 | 6 | 9 |
| MVFL | 1.22 | 1.21 | 1.24 | 2.14 | 2.14 | 2.15 | 4.03 | 4.10 | 4.02 | 0.20 | 0.23 | 0.25 |
| VFL | 1.29 | 1.31 | 1.30 | 2.27 | 2.28 | 2.29 | 4.23 | 4.29 | 4.19 | 0.30 | 0.36 | 0.35 |

| Datasets | ETTm | | | ETTh | | | weather | | | exchange | | |
|---|---|---|---|---|---|---|---|---|---|---|---|---|
| exchange size | 3 | 6 | 9 | 3 | 6 | 9 | 3 | 6 | 9 | 3 | 6 | 9 |
| MVFL | 1.75 | 1.72 | 1.70 | 2.41 | 2.39 | 2.39 | 3.83 | 3.84 | 3.85 | 0.19 | 0.20 | 0.24 |
| VFL | 1.78 | 1.74 | 1.75 | 2.55 | 2.50 | 2.49 | 3.88 | 3.87 | 3.89 | 0.32 | 0.31 | 0.31 |

VFL approach, MVFL gives a loss that is **64%** of that of VFL with only **57%** of the communication resources usage. In situations where the storage resources are severely limited, MVFL could still reduce the loss to **81%** of that of VFL with only **21%** of the storage resources usage under the input-96 setting for the exchange dataset. Overall, MVFL yields a loss that is 87.9% of that of VFL with a 83% storage resources usage and a 57% communication resources usage. Or in more extreme situations, MVFL could maintain the loss to 96% of that of VFL with only 25% storage resources usage and 57% communication resources usage.

### 5.4 RESULTS ANALYSIS

The benefits of MVFL over VFL is firstly related to the number of devices involved in the process. In a general case, VFL would require $n-1$ times more storage resources usage compared to MVFL. The more devices there are, the more storage resources are saved by MVFL.

In addition, the exchange models of VFL take the same input size of the main local model. This means that the exchange model size $m$ increase with the input size. As a result, MVFL is more advantageous when the input size is large. This could be verified by the experiment results: The storage results could be saved more when input size is 96 than when input size is 24 (case MVFL2), and if we force the storage resources usage of MVFL to approach the storage resources usage of VFL (case MVFL1), the loss of MVFL under input size 96 is much more advantageous.

### 5.5 ABLATION STUDIES

In order to study the impact of different communication feature sizes over the performances of MVFL and VFL, we fix the input size and set communication features sizes to 3, 6, 9. Same storage usage are set for MVFL and VFL. The results are shown in the Tables 2. We find no clear patterns indicating whether the performance of MVFL relative to VFL improves or deteriorates as

Table 3: Loss of CMVFL and MVFL over different exchange sizes under the input-24 setting

| Datasets | ETTm | | | ETTh | | | weather | | | exchange | | |
|---|---|---|---|---|---|---|---|---|---|---|---|---|
| exchange size | 3 | 6 | 9 | 3 | 6 | 9 | 3 | 6 | 9 | 3 | 6 | 9 |
| CMVFL | 1.88 | 1.87 | 1.85 | 2.60 | 2.63 | 2.62 | 4.15 | 4.13 | 4.13 | 0.21 | 0.23 | 0.25 |
| MVFL | 1.75 | 1.72 | 1.70 | 2.41 | 2.39 | 2.39 | 3.83 | 3.84 | 3.85 | 0.19 | 0.20 | 0.24 |

Table 4: Loss of SMVFL and MVFL over different exchange sizes under the input-24 setting

| Datasets | ETTm | | | ETTh | | | weather | | | exchange | | |
|---|---|---|---|---|---|---|---|---|---|---|---|---|
| exchange size | 3 | 6 | 9 | 3 | 6 | 9 | 3 | 6 | 9 | 3 | 6 | 9 |
| SMVFL | 1.87 | 1.82 | 1.77 | 2.56 | 2.51 | 2.51 | 3.95 | 3.93 | 3.88 | 0.26 | 0.25 | 0.31 |
| MVFL | 1.75 | 1.72 | 1.70 | 2.41 | 2.39 | 2.39 | 3.83 | 3.84 | 3.85 | 0.19 | 0.20 | 0.24 |

the exchange size increases. However we observe that for a certain dataset under a same input setting, for both MVFL and VFL, a common exchange size performs better. For example, for weather dataset under the input-96 setting, a best exchange size is 9 for both MVFL and VFL. This could be relevant to the different mutual correlation of different datasets.

We have conducted another set of ablation studies with a more compacted version of MVFL, which we refer to as compact MVFL (CMVFL). In CMVFL, the communication features from different devices are not concatenated, but rather averaged. As a result, the model size of CMVFL would always be constant, no matter the exchange size or the total devices number. Comparison results between MVFL and CMVFL under the same model settings are as shown in Table 3.

The results show that MVFL is advantageous over CMVFL in all cases, indicating that merging the communication features from all devices does not perform well. This is perhaps due to the fact that the information extracted from different variates of the dataset could be very different. Consequently, information loss could happen when we force the merging of the communication features.

The last set of ablation studies discusses another version of MVFL, which we refer to as simple MVFL (SMVFL). In SMVFL, the communication features are sent to the server, but devices wouldn't send the gradients for other communication features to the server, nor would them receive the feedback gradients of their own communication features. Consequently, the only difference between communication features and local features would be their relative position in the embedded feature space. The comparison results of SMVFL and VFL under the same model settings are as shown in Table 4.

The results show that MVFL is advantageous over SMVFL in all cases. This further proves that the MVFL framework is valid and the approach *a posteriori* to separate communication features and local features could indeed improve the performance of the framework.

## 6 CONCLUSIONS

In this work, we proposed a novel framework, Multivariate Vertical Federated Learning (MVFL) to address the challenges of resource-efficient multivariate time series forecasting on resource-constrained IoT devices. By revisiting traditional Vertical Federated Learning (VFL) in the targeted scenario where each device must perform local forecasting with vertically partitioned data, We identified its limitations with high storage space on local devices. Our proposed MVFL framework separates communication features and local features in an embedded space, offering a more efficient approach with better performances. This also prevents a rapid linear increase of model size with the increase of devices. Future work may explore the adaptation of MVFL to more sophisticated model structures or existing resource-efficient approaches and further optimize its performance under diverse conditions.

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
