# OpenReview forum: "MVFL: Multivariate Vertical Federated Learning for Time-Series Forecasting"
_ICLR.cc/2025/Conference — Submitted to ICLR 2025_

### Official Review · Reviewer_MgTm · 2024-10-29

**Soundness:** 2
**Presentation:** 3
**Contribution:** 2
**Rating:** 6
**Confidence:** 3

**Summary:**

This paper proposes Multivariate Vertical Federated Learning (MVFL) to address the challenges of resource-efficient multivariate time series forecasting on resource-constrained IoT devices. By revisiting traditional Vertical Federated Learning (VFL) in the targeted scenario where each device must perform local forecasting with vertically partitioned data, this paper identified its limitations with high storage space on local devices. The proposed MVFL framework separates communication features and local features in an embedded space, offering a more efficient approach with better performances. This also prevents a rapid linear increase of model size with the increase of devices.

**Strengths:**

1.	The structure of this paper is clear and reasonable.
2.	This paper is easy to follow.

**Weaknesses:**

1.	The literature review of this paper is a little limited. The literature review would be enhanced to include multi-modal FL methods as well as efficient FL methods.
2.	The FL methods associated with multimodality also seem to be applicable to the case where the variables are distributed across different clients. It would be useful to have a discussion of these methods and to describe the differences and advantages of the proposed methods.
3.	It is not clear how to segment the communication and local features. Despite Section 4.3 attempts to explain the segmentation of the communication and local features, however, the specifics of this section focus on how to calculate the gradients of the local and communication features, rather than segmentation.
4.	The baselines used in the experiment is not sufficiently representative of the SOTA approach. It would be better to include more fresh and relevant work in the comparison experiments, especially in support of efficient VFL methods that deal with multiple variables.

**Questions:**

What are the differences and connections between “multivariate” VFL and “multimodal” VFL?

---

> ### Author Response · Authors · 2024-11-20
>
> Dear reviewer, thanks for your efforts and the insights! About the weaknesses you have pointed out, we will discuss them in detail.
>
> 1.About point 1 in the weakness part: “The literature review of this paper is a little limited. The literature review would be enhanced to include multi-modal FL methods as well as efficient FL methods.”
>
> It is a valuable suggestion and we agree that multi-modal FL models could also be helpful to provide insights about vertically distributed data and we have added multi-modal FL researches in the second paragraph of section 2.1.
>
> 2.About point 2 in the weakness part: “The FL methods associated with multimodality also seem to be applicable to the case where the variables are distributed across different clients. It would be useful to have a discussion of these methods and to describe the differences and advantages of the proposed methods.”
>
> It is a very valuable suggestion and we added a discussion part in the second paragraph of section 2.1. The core idea is as follows: we think that multi-modal FL methods could not be applied directly to our targeted scenario because for FL methods (according to the context, we believe that here you mean horizontal FL), exchanges between clients only happens at the training stage. However, in our scenario, even after all the models have converged, if a client would like to make predictions on its own variate, it would still need the communication features from other clients, because a basic assumption is that the real-time data (not a static behavior mode) of one client could contain useful information to help the prediction of other clients’ data.
>
> If you mean multimodal VFL, please refer to point 5 of this reply, which addresses how MVFL could also be useful for multimodal VFL.
>
> 3.About point 3 in the weakness section: “It is not clear how to segment the communication and local features.”
>
> Thanks for pointing out the unclarity! In fact, our segmentation method is not a method a priori (meaning we come up with some criteria to decide which neurons could be sent to others and which neurons should stay local), but rather a method a posteriori (meaning that at the very beginning of the training stage, you cannot tell communication features from local features, for they become what they are as the training process goes). This means that we arbitrarily let the first several neurons (literally the first several, meaning that here we only consider the relative location) be the communication features. As the training goes, we calculate the gradients for those neurons using the communication features’ gradients while for other neurons, we use the local features’ gradients. Consequently, when the training process ends, we could get the real communication features as indicated by the name. We could make a simple yet illustrative and intuitive analogy to understand this process: imagine that in the 19th century of the British Empire, a pair of twin brothers were adopted by two different families: one by an upper-class family and the other by a working-class family. By the time they are raised, the one adopted by the upper class will look like an upper-class person and the other will look like a laborer.
>
> 4.About the point 4 in the weakness section: “The baselines used in the experiment is not sufficiently representative of the SOTA approach.”
>
> To our best knowledge, our work is the first to tackle the targeted scenario, where the variates are vertically distributed and each device needs to do local forecasting. The comparison experiment has been performed between VFL and MVFL because MVFL is a variant of VFL specially designed for the case where each device needs to do local forecasting: The problem of VFL that we are addressing is universal for all VFL-based approaches, since the redundant models would take up too much resources if all parties are active parties that need to perform local forecasting.
>
> 5.About the question on the difference between multivariate and multimodal VFL, we think that in the context of VFL, they are very similar because they both require information sharing during not only the training stage, but also the prediction/classification/regression stage and the data on a client is significantly different from another client since data are vertically distributed. One of the differences could be the tasks: while the multivariate time series dataset often focuses on forecasting tasks, the multimodal dataset could involve a broader range of tasks such as classification and conversion. Nevertheless, for both “multivariate” VFL and “multimodal” VFL, there are often only a limited proportion of devices that are active parties and both approaches could be improved by MVFL if all parties need to do local forecasting.
>
> We thank you again for you time and efforts and see to your replies.

---

> > ### Comment · Reviewer_MgTm · 2024-11-28
> >
> > Thanks for the authors' responses to my review.
> >
> > However, some issues are still not very clear.
> > About Q3: It's still not clear what's the exact method to segment the communication and local features. Althogh the authors highlight that they used a "posteriori" to realize such segment, as well as provide a vivid example about "twin brothers." It is noted that when the number of features reaches the level of thousands or even millions, the criteria of "look like" used in the "twin brothers" example seems not suitable for this work.
> > About Q4: I didn't see improvement on comparison experiments. Many VFL methods have been proposed in previous studies. However, the authors only used the classical VFL method as the only baseline, which is not sufficient to provide convincing comparative results.
> >
> > So, I will maintain my score.

---

> > > ### Author Response · Authors · 2024-11-29
> > >
> > > Dear reviewer, thanks for the reply. We will try to further clarify your concerns.
> > >
> > > About Q3, the segmentation a posteriori means we don’t segment communication features and local features manually. Rather, we train them using different gradient strategies so that they converge differently. From your reply, we think that perhaps your concern is on how we can be sure that this segmentation method is valid. You could refer to our comparison experiments between SMVFL and MVFL in the Table 4 of the Ablation studies section. In SMVFL, the communication features are sent to the server, but devices wouldn’t send the gradients for other communication features to the server, nor would them receive the feedback gradients of their own communication features. Consequently, there would be no difference between “communication features” and local features. As the results show, SMVFL perform worse than MVFL on all benchmarks and this could validate this segmentation method.
> > >
> > > About Q4, the improvement could be observed from two comparisons, as we have put it in our abstract and main results section: the comparison between MVFL1 and VFL shows that when the storage resources are equally utilized, MVFL yields a 12.1\% relative improvement on loss; the comparison between MVFL2 and VFL shows that Even when both MVFL and VFL employ the same main model size, MVFL achieves a 75\% reduction in storage resources compared to VFL while maintaining the loss at the same level of VFL. It is true that many VFL methods have been proposed in previous studies, as we have also put them in the related works section. However, for the targeted scenario, MVFL is the first to address the redundancy problem of VFL (which is a common problem for all VFL-based methods) that we have pointed out in section 3, thus we expect it to be helpful for all VFL-based approaches as long as its advantages could be validated when comparing to classical VFL.

---

### Official Review · Reviewer_n5RJ · 2024-11-03

**Soundness:** 3
**Presentation:** 3
**Contribution:** 4
**Rating:** 6
**Confidence:** 4

**Summary:**

The paper introduces MVFL, a novel federated learning framework for multivariate time-series forecasting on resource-limited devices. It addresses the inefficiency of traditional VFL by separating communication and local features in an embedded space, reducing storage and communication costs. The methodology involves a modified VFL approach with a focus on efficient resource use, demonstrated through experiments on real-world benchmarks.

**Strengths:**

1. Originality:
The paper presents a novel approach to federated learning that is tailored for time-series forecasting, which is a creative extension of existing VFL frameworks.
2. Quality:
The experiments are well-designed and demonstrate the effectiveness of MVFL in terms of efficiency and accuracy.
3. Clarity:
The paper is clearly written, with a logical flow that makes the complex concepts accessible.
4. Significance:
The work addresses a significant problem in IoT and federated learning, with implications for practical applications. It also provide insight for possible future more efficient multivariate time series forecasting solutions.

**Weaknesses:**

While this work presents a novel method with significant contributions to the field of multivariate time-series forecasting on resource-limited devices, there are areas where the work could be improved:

Though the novelty of the proposed model is promising, its applicability on real-world datasets could be further validated with a broader range of diverse datasets. This work aims to address the challenges of multivariate time series forecasting; however, the datasets used do not adequately demonstrate its effectiveness in this regard. The largest dataset they used, Weather, includes only 21 variables, whereas real-world time series data often involves far more complex variables, as seen in datasets such as Traffic (862 variables), ECL (321 variables), and PEMS07 (883 variables).

Additionally, have the authors considered evaluation metrics other than MSE and MAE? The R-squared (the coefficient of determination) is a quite commonly used evaluation metric in time-series forecasting. While MSE is valuable for understanding prediction errors, R-squared provides a broader view of how well the model captures the underlying relationships in the data.

**Questions:**

1. Are there any real-world application scenarios where MVFL has been tested or is planned to be tested？The authors could discuss the potential challenges and benefits of deploying MVFL in practical IoT environments.

2. How sensitive is the performance of MVFL to the choice of forecasting model? Would the authors share any insights or best practices ?

3. Besides, the authors could investigate the use of interpretable forecasting paradigms like N-BEATS, or decomposition strategies such as Seasonal and Trend decomposition used in Autoformer and Leddam, to enhance the transparency of their model and provide insights into the patterns learned from data.

---

> ### Author Response · Authors · 2024-11-20
>
> Dear reviewer, thanks for your efforts and the insights! We will further discuss the weaknesses that you have pointed out and try to answer the questions you have posed.
>
> 1.For the first point you have pointed out in the weakness part: “Though the novelty of the proposed model is promising, its applicability on real-world datasets could be further validated with a broader range of diverse datasets.”
>
> We agree that the number of variables could be increased. However, in many multivariate datasets, for example the datasets you have mentioned, the variates could not be considered as vertically distributed.
>
> Particularly, Traffic [3] dataset records time series data from traffic sensors, typically used for traffic flow forecasting tasks. Each variable corresponds to a specific sensor, capturing the traffic conditions at its location. ECL [4] tracks electricity consumption of multiple users over time. It is widely used for energy load forecasting and analyzing seasonal trends in electricity usage. PEMS07[5] is part of the California Performance Measurement System, containing traffic flow data from 883 sensors on California highways. It is often used to benchmark traffic forecasting models.
>
> It is true that, for example, the traffic density of one cross could somehow affect the traffic density of another cross, but they are both traffic density data and could have shared patterns and behavior modes. In this case, we can not be sure about whether approaches based on VFL (like ours) could be advantageous over approaches based on horizontal FL. If we use the same example, the kind of data that we want is not traffic density on different crosses, but the traffic density, noise level, weather condition etc. on only one cross. Of course, it would be very interesting to integrate all those variates also on different crosses but that would be another scenario. A good solution for that scenario, for example, could be a method to somehow combine the VFL and HFL approaches and we believe that our MVFL could still be useful for that hybrid approach since it is a variant of the classical VFL method.
>
> 2.For the second weakness you have pointed out: “Additionally, have the authors considered evaluation metrics other than MSE and MAE? The R-squared (the coefficient of determination) is a quite commonly used evaluation metric in time-series forecasting.”
>
> We admit that we have followed the metrics seen in some papers like Autoformer [8], and have not considered other evaluation metrics. We have done experiments to show the advantage of our method for the new metric of R-squared. We have included the results in the Appendix A.1 based on your suggestions. Concretely, when we train with L2 loss and evaluate with R2 metric, the advantages of MVFL over VFL still persist.
>
> 3.About the question1: “The authors could discuss the potential challenges and benefits of deploying MVFL in practical IoT environments.”
>
> We believe that the targeted scenario exists in different fields. A common example could be in the field of smart cities [1]. In many smart cities, data from traffic flow, air quality, temperature, and noise levels are collected at different intersections or city sectors by separate sensor devices. Each device typically monitors one specific type of data due to cost constraints or location-specific deployment. Their abilities to forecast future data for own variables could be significantly improved if they adapt the MVFL method: Apart from the security and privacy issues, a great advantage of MVFL is that it enables each device to communicate with other devices in a communication-efficient way, for it could, to some extent, “compress” the past data into the communication features and send it to other devices, which is way more efficient than a centralized approach. The challenges are as discussed in the reply to the first weakness you have pointed out: For large multivariate datasets where the variates are not vertically distributed, but rather horizontally distributed, a novel approach combining VFL and FL should be designed.
>
> There are real-world scenarios that could be found in financial forecasting across banking sectors, where customer data from one branch cannot be shared directly with another due to privacy regulations like GDPR [4], or in the field of healthcare wearables [5][6][7] or chronic disease monitoring, since health data from wearables is highly sensitive.
>
> Sorry that we have reached the characters limitation for this reply. We will include the rest in the next one.

---

> > ### Author Response · Authors · 2024-11-20
> >
> > 4.About the question2: “How sensitive is the performance of MVFL to the choice of forecasting model? Would the authors share any insights or best practices?”
> >
> > A common phenomenon is that the larger the model size, the better the performance. Of course, we are in a context of resource-limited devices where the model sizes could not be very large, so we have chosen a humble size for the models (around hundreds of KB, as can be seen in Table1). For the same reasons, the trendy transformer-based models are not suitable for the targeted scenario due to their complexity.
> >
> > 5.About the question3: “the authors could investigate the use of interpretable forecasting paradigms like N-BEATS, or decomposition strategies such as Seasonal and Trend decomposition used in Autoformer and Leddam”
> >
> > It is a very valuable suggestion. Also, the works you have mentioned are interesting works and we have added them in the first paragraph of section 2.2 accordingly. The adopted model is a simple multilayer perceptron, without the trendy transformer-based mechanisms and other mechanisms focusing on decomposition like N-BEATS [11], for we are aiming at resource-limited devices thus the simpler the model, the better. Besides, according to [2], in our settings, the trendy models, for example Autoformer and which includes the transformer and decomposition mechanisms, do not show significant advantages over simpler approaches.
> >
> > However, we agree with your insights that it would be better that we add other experiments using other paradigms. We are trying to add another experiment using Informer [9]. As a foundational work for Autoformer, the Informer model introduces the transformer mechanisms for time series forecasting. We are now trying to adapt the MVFL approach for the Informer by taking the outputs of the encoder as the embedded layer where we segment the communication features and the local features.
> >
> > As for Autoformer and Leddam [10], since they introduce the decomposition mechanisms of trend and seasonality, we could adapt the MVFL to them by executing the communication features exchanges separately for the trend models and the seasonality models. We expect to get the results if time allows. Otherwise, we could include those in a future work.
> >
> > We thank you again for you time and efforts and see to your replies.
> >
> > [1] Pandya S, Srivastava G, Jhaveri R, et al. Federated learning for smart cities: A comprehensive survey[J]. Sustainable Energy Technologies and Assessments, 2023, 55: 102987.
> >
> > [2] Zeng A, Chen M, Zhang L, et al. Are transformers effective for time series forecasting?[C]//Proceedings of the AAAI conference on artificial intelligence. 2023, 37(9): 11121-11128.
> >
> > [3] A. Wang, Y. Ye, X. Song, S. Zhang and J. J. Q. Yu, "Traffic Prediction With Missing Data: A Multi-Task Learning Approach," in IEEE Transactions on Intelligent Transportation Systems, vol. 24, no. 4, pp. 4189-4202, April 2023, doi: 10.1109/TITS.2022.3233890. keywords: {Task analysis;Predictive models;Training;Multitasking;Feature extraction;Deep learning;Data mining;Traffic speed prediction;missing data;spatio-temporal modeling;deep learning;multi-task learning},
> >
> > [4]Voigt P, Von dem Bussche A. The eu general data protection regulation (gdpr)[J]. A Practical Guide, 1st Ed., Cham: Springer International Publishing, 2017, 10(3152676): 10-5555.
> >
> > [5] Rieke N, Hancox J, Li W, et al. The future of digital health with federated learning[J]. NPJ digital medicine, 2020, 3(1): 1-7.
> >
> > [6] Wang J, Chen Y. Federated Learning for Personalized Healthcare[M]//Introduction to Transfer Learning: Algorithms and Practice. Singapore: Springer Nature Singapore, 2022: 303-313.
> >
> > [7] Xu J, Glicksberg B S, Su C, et al. Federated learning for healthcare informatics[J]. Journal of healthcare informatics research, 2021, 5: 1-19.
> >
> > [8] Wu H, Xu J, Wang J, et al. Autoformer: Decomposition transformers with auto-correlation for long-term series forecasting[J]. Advances in neural information processing systems, 2021, 34: 22419-22430.
> >
> > [9] Zhou H, Zhang S, Peng J, et al. Informer: Beyond efficient transformer for long sequence time-series forecasting[C]//Proceedings of the AAAI conference on artificial intelligence. 2021, 35(12): 11106-11115.
> >
> > [10] Yu G, Zou J, Hu X, et al. Revitalizing multivariate time series forecasting: Learnable decomposition with inter-series dependencies and intra-series variations modeling[J]. arXiv preprint arXiv:2402.12694, 2024.
> >
> > [11] Oreshkin B N, Carpov D, Chapados N, et al. N-BEATS: Neural basis expansion analysis for interpretable time series forecasting[J]. arXiv preprint arXiv:1905.10437, 2019.

---

> > > ### Comment · Reviewer_n5RJ · 2024-11-25
> > >
> > > Dear Authors,
> > >
> > > Thank you for your thorough responses to my review. I appreciate the clarifications you've provided, which have helped to address most of my questions.
> > >
> > > While I still have concerns about whether your method can represent real-world environments effectively, especially with a larger number of devices or more complex algorithms, I acknowledge the novelty and potential of your approach. The scenarios you address may become more prevalent as IoT devices continue to proliferate. So, my rating remains the same.

---

### Official Review · Reviewer_yHdr · 2024-11-04

**Soundness:** 3
**Presentation:** 3
**Contribution:** 2
**Rating:** 5
**Confidence:** 4

**Summary:**

The authors propose an approach MVFL to forecast the time series in resource-constraint IoT scenarios. They separated communication features and local features in the embedded feature space and used storage and communication resources more efficiently by eliminating redundant models. Experiments on efficiency and accuracy are conducted to evaluate the performance of the proposed mechanism.

**Strengths:**

1. The considered scenario is novel. Considering limited resources in federated learning for time-series analysis is significant.
2. Experiments show that the performance of the proposed method is significantly better than existing methods in different scenarios.
3. The paper is generally well presented.

**Weaknesses:**

1. Traditional VFL has different label spaces between clients. The time-series forecasting proposed in this article is not a classification task, so is it a reasonable scenario for each client to hold the features of one dimension in a multi-dimensional time-series? Will there be such an application in reality? In addition to the weather dataset, will other common time-series forecasting datasets such as Electricity, Traffic, and ILI also exhibit such a scenario? It seems to be a synthetic scenario.

2. How can privacy issues of communication features be preserved?

3. The adopted model is unclear. What about the trendy transformer-based model or other time series forecasting models, like PatchTST, TimesNet, FEDformer, Informer or others?

4. The comparisons in experiments are insufficient. The proposed methods are only compared with VFL. It can be compared with other FL for time-series forecasting like MetePFL[1] or VFL work (changing the downstream task). Only one baseline cannot validate the effectiveness of the proposed method.

[1] Chen, Shengchao, et al. "Prompt federated learning for weather forecasting: toward foundation models on meteorological data." in IJCAI 2023.

**Questions:**

Refer to weakness.

---

> ### Author Response · Authors · 2024-11-20
>
> Dear reviewer, thanks for your efforts and the insights! About the points you have pointed out, we will discuss them one by one.
>
> 1.About point 1: “Will there be such an application in reality? In addition to the weather dataset, will other common time-series forecasting datasets such as Electricity, Traffic, and ILI also exhibit such a scenario? It seems to be a synthetic scenario.”
>
> A common example could be in the field of smart cities [1]. In many smart cities, data from traffic flow, air quality, temperature, and noise levels are collected at different intersections or city sectors by separate sensor devices. Each device typically monitors one specific type of data due to cost constraints or location-specific deployment. Their abilities to forecast future data for own variables could be significantly improved if they adapt the MVFL method.
>
> Other real-world scenarios could be found in financial forecasting across banking sectors, where customer data from one branch cannot be shared directly with another due to privacy regulations like GDPR [5], or in the field of healthcare wearables for chronic disease monitoring [6][7][8], since health data from wearables is highly sensitive. Therefore, we believe that the targeted scenario is not a synthetic scenario.
>
> 2.About point 2: “How can privacy issues of communication features be preserved?”
>
> the proposed MVFL method is a variant of the classical VFL method. To be specific, in VFL practices, all the neuron values of a final embedded layer are communicated while in MVFL, only a portion of those neuron values (the communication features) are communicated. This implies that the privacy could be preserved for MVFL if the privacy could be preserved for VFL. Consequently, the privacy issues are not heavily discussed in our paper. About the privacy issues concerning VFL, you could refer to [2].
>
> Besides, when we compare the security performance of VFL-based approaches (such as MVFL) and FL-based approaches, we observe that in the context of federated learning [9][10], a common privacy concern arises from the possibility of model reconstruction attacks. However, with our proposed method, the value of neurons from a specific layer alone cannot reveal the models of individual clients, and this intuitively offers an additional advantage in terms of privacy.
>
> 3.About point 3: “The adopted model is unclear. What about the trendy transformer-based model or other time series forecasting models, like PatchTST, TimesNet, FEDformer, Informer or others?”
>
> It is a valuable question. The adopted model is a simple multilayer neural network, without the trendy transformer-based mechanisms. This choice could be justified by two points: Firstly, according to [3], in our settings, the trendy transformer models, for example Autoformer[11] and Informer[12], do not show significant advantages over simpler approaches, so a simple MLP is enough to show the advantages of MVFL. Secondly, since we are aiming at resource-limited devices, it is better to consider lighter models instead of the heavy transformer-based models.
>
> However, we agree with your insights that it would be better that we add another experiment using transformer-based models in order to show the robustness of our method. We are currently working on Informer. Concretely, we are going to take the outputs of the encoder as the embedded layer where we segment the communication features and the local features. We are confident that this approach would greatly enhance the performance and we expect to get the results for the rebuttal if time allows. We would tackle this case in a future work otherwise.
>
> Sorry that we have reached the characters limitation for this reply, we would include other points and citations in the next one.

---

> ### Author Response · Authors · 2024-11-20
>
> 4.About point 4: “It can be compared with other FL for time-series forecasting like MetePFL or VFL work (changing the downstream task). Only one baseline cannot validate the effectiveness of the proposed method.”
>
> We have checked the paper by Chen, Shengchao, et al [13], which you have mentioned. It is a relevant work addressing time series forecasting, but the research does not tackle our targeted scenario for two reasons. Firstly, in [13], the data are not vertically distributed and the goal is to train an integral model to do forecasting for all variates (all located on a client) in a centralized way and with a pretraining stage, whereas in MVFL, each device has a local model to forecast the future data only for its own variate (it does not have direct access to other variates). Secondly, a basic assumption of the targeted scenario is that the real-time data (not a static behavior mode) of one client could contain useful information to help the prediction of other clients’ data. Consequently, even after all the models have converged, if a client would like to make predictions on its own variate, it would still need the communication features from other clients. However, in [13], it seems that the exchange between clients (through the server) only happens at the training stage. We believe that the classical VFL approach that we have chosen is representative enough to show the advantages of MVFL, since MVFL tackles the problem of redundant models for all VFL-based approaches. Meanwhile, we agree that we could add more papers like this in our related works section and we have done so in the first paragraph of section 2.2.
>
> We thank you again for you time and efforts and see to your replies.
>
> [1] Pandya S, Srivastava G, Jhaveri R, et al. Federated learning for smart cities: A comprehensive survey[J]. Sustainable Energy Technologies and Assessments, 2023, 55: 102987.
>
> [2] Liu Y, Kang Y, Zou T, et al. Vertical federated learning: Concepts, advances, and challenges[J]. IEEE Transactions on Knowledge and Data Engineering, 2024.
>
> [3] Zeng A, Chen M, Zhang L, et al. Are transformers effective for time series forecasting?[C]//Proceedings of the AAAI conference on artificial intelligence. 2023, 37(9): 11121-11128.
>
> [4] Yang Liu, Yan Kang, Tianyuan Zou, Yanhong Pu, Yuanqin He, Xiaozhou Ye, Ye Ouyang, Ya-Qin Zhang, and Qiang Yang. Vertical federated learning: Concepts, advances, and challenges. IEEE Transactions on Knowledge and Data Engineering, 2024.
>
> [5]Voigt P, Von dem Bussche A. The eu general data protection regulation (gdpr)[J]. A Practical Guide, 1st Ed., Cham: Springer International Publishing, 2017, 10(3152676): 10-5555.
>
> [6] Rieke N, Hancox J, Li W, et al. The future of digital health with federated learning[J]. NPJ digital medicine, 2020, 3(1): 1-7.
>
> [7] Wang J, Chen Y. Federated Learning for Personalized Healthcare[M]//Introduction to Transfer Learning: Algorithms and Practice. Singapore: Springer Nature Singapore, 2022: 303-313.
>
> [8] Xu J, Glicksberg B S, Su C, et al. Federated learning for healthcare informatics[J]. Journal of healthcare informatics research, 2021, 5: 1-19.
>
> [9] Mothukuri V, Parizi R M, Pouriyeh S, et al. A survey on security and privacy of federated learning[J]. Future Generation Computer Systems, 2021, 115: 619-640.
>
> [10] Lyu L, Yu H, Yang Q. Threats to federated learning: A survey[J]. arXiv preprint arXiv:2003.02133, 2020.
>
> [11] Wu H, Xu J, Wang J, et al. Autoformer: Decomposition transformers with auto-correlation for long-term series forecasting[J]. Advances in neural information processing systems, 2021, 34: 22419-22430.
>
> [12] Zhou H, Zhang S, Peng J, et al. Informer: Beyond efficient transformer for long sequence time-series forecasting[C]//Proceedings of the AAAI conference on artificial intelligence. 2021, 35(12): 11106-11115.
>
> [13] Chen, Shengchao, et al. "Prompt federated learning for weather forecasting: toward foundation models on meteorological data." in IJCAI 2023.

---

> ### Comment · Reviewer_yHdr · 2024-11-25
>
> Thanks for your answers. I still have some concerns.
>
> Q1: For the given examples of smart cities in your response, the “variate” for time series becomes “domains” as traffic, air, and temperature data. Generally, multivariate time series seems to be in-domains as in your paper’s example: data of temperature, wind, and h2o density, all in the weather domain. For data within the domain, I think this data may be held entirely by one institution, such as different monitoring devices owned by a weather station, rather than in a scenario where federated learning is required between banks. So, my question is, in your MVFL, if the variates of a time series are in one domain, it seems that their ownership should be consistent and no federated learning is needed. If they are in different domains, this seems to be more of a cross-domain federated learning problem like [1], rather than a multivariate problem in the time series itself.
>
> Q3:There are also many non-transformer-based models, such as convolution-based TimesNet (ICLR 23), MICN (ICLR 23), SCINet (NeurIPS 22),  and MLP-based: DLinear (AAAI 23), N-HITS(AAAI 23), N-BEATS(ICLR 20) and many others. The current version of the paper only mentions how many hidden layers. I have no idea about the specific details and sources of your model, let alone how it compares to the transformer, CNN and MLP time series models. My original question was not about computing costs, since convolutional and MLP-based models generally have lower consumption than transformer-based ones.
>
> Q4: Although MetePFL is not specifically designed for MVFL as you analyzed, since there is relatively little work on federated time-series learning or federated weather forecasting, I think it is worth comparing it with the same data partitioning. There is also a lot of work on VFL, and it is worth comparing them according to your data partitioning. I think having only one baseline does not prove the effectiveness of your method.
>
> [1] Liu Q, Liu X, Liu C, et al. Time-FFM: Towards LM-Empowered Federated Foundation Model for Time Series Forecasting. NeurIPS 2024.

---

> > ### Author Response · Authors · 2024-11-28
> >
> > Dear reviewer, thanks for the reply. We will further discuss the questions.
> >
> > About Q1, in fact, we have never limited the scenario of our research to be “in one domain” or “cross-domain”. In the one-domain scenario, the main advantage of MVFL is its communication efficiency: we don’t need to communicate all the raw data, but rather the communication features as kind of a “compressed data”. About the cross-domain scenario, we have checked the Time-FFM research you have cited. However, it is still a horizontal FL work and do not apply to our targeted scenario for the same reasons as Mete-PFL.
> >
> > About Q3, first, please accept our gratitude for pointing out so many SOTA works on time series forecasting. While we were trying our best modifying those works in order to obtain a VFL variant and an MVFL variant, we have identified a problem in our original experiments, which we couldn’t have noticed without diving deeply into those works: In time series forecasting, a naïve random train-test split might cause data leakage [1]. We have redone our experiments using the proper data splitting method and updated the results in the pdf document. We are lucky that the conclusions still hold and the advantage of MVFL over VFL becomes even more obvious.
> > About the model we are using, it is a very simple MLP with 3 hidden layers, where the layer in the middle is used as the embedded feature space where we separate communication features and local features. We have tried added other mechanisms such as decomposition and dropout etc., but it proves that this simple model works best on both MVFL and VFL. The sizes of the models are indicated in the main results table of the pdf document.
> > We are still struggling to provide you with the experiments using a different SOTA time series model. The major difficulty is that those are all well-capsulated works. It is relatively easy to use them directly, but to modify them in order to get an MVFL version and a VFL version requires a lot of hard work.
> > However, we would like to emphasize that MVFL is not a new SOTA work for time series forecasting like Autoformer, Informer, NBeats or MetePFL, Time-FFM etc. and that we only need to prove the advantages of MVFL over VFL with an illustrative model like ours in order to validate our work. As we have put it in our Conclusion section, we acknowledge that the comparison experiments using other models could further prove the robustness of MVFL and we could include them in future research.
> >
> > About Q4, we have to point out that the core idea of this paper is to provide a better framework of VFL for the scenario of multivariate forecasting. MVFL is not a new SOTA work on federated weather forecasting and it is not necessary to compare it with existing SOTA works like MetePFL (in MetePFL, the FL framework means the exchange between devices only happens at training stage, but not forecasting stage. This directly negates a basic assumption of the targeted scenario: the real-time data instead of a static behavior mode of one client could contain useful information to help the prediction of other clients’ data. Or we could put it more simply: At the forecasting stage, a device of MetePFL is simply doing univariate forecasting). As we have put it in the related works section, to our best knowledge, MVFL is a gap-addressing work, meaning that MVFL could be helpful to whatever VFL-based method for the targeted scenario, for none of those methods address the redundancy problem that we have pointed out. Therefore, we insist that to compare with classical VFL is enough to show the advantages of MVFL.
> >
> > [1] Hyndman R J. Forecasting: principles and practice[M]. OTexts, 2018.

---

### Comment · Area_Chair_y5rW · 2024-11-25
**Acknowledge the author responses**

Dear Reviewers,

Thank you very much for your effort. As the discussion period is coming to an end, please acknowledge the author responses and adjust the rating if necessary.

Sincerely,
AC

---

### Comment · Area_Chair_y5rW · 2024-11-28
**Discussion needed**

Dear Reviewers,

As you are aware, the discussion period has been extended until December 2. Therefore, I strongly urge you to participate in the discussion as soon as possible if you have not yet had the opportunity to read the authors' response and engage in a discussion with them. Thank you very much.

Sincerely,
Area Chair

---

### Meta-Review · Area_Chair_y5rW · 2024-12-19

**Metareview:**

This paper studies a resource-efficient multivariate time-series forecasting based on vertical federated learning (VFL).  Although the paper presents an interesting, important application scenario, the reviewers agreed that the merit of the proposed method is not fully proven through convincing evaluation.  Multiple reviewers pointed out that the novelty is not convincingly presented.  Thus, I would like to recommend a reject.

**Additional Comments On Reviewer Discussion:**

* Reviewer n5RJ pointed out that the datasets utilized in the authors' experiments may not capture the full complexity of the scenarios they intend to address, which in turn raises questions about the practical effectiveness of their methodology in real-world settings.

* Reviewers yHdr and n5RJ agreed upon the lack of the baselines for evaluation.

Thus, we reached a consensus that the paper is not ready for publication.

---

### Decision · Program_Chairs · 2025-01-22

Reject